# Therapeutic and Diagnostic Potential of Folic Acid Receptors and Glycosylphosphatidylinositol (GPI) Transamidase in Prostate Cancer

**DOI:** 10.3390/cancers16112008

**Published:** 2024-05-25

**Authors:** Marco Hoffmann, Thomas Frank Ermler, Felix Hoffmann, Radu Alexa, Jennifer Kranz, Nathalie Steinke, Sophie Leypold, Nadine Therese Gaisa, Matthias Saar

**Affiliations:** 1Department of Urology and Pediatric Urology, University Medical Center RWTH Aachen, 52074 Aachen, Germany; termler@ukaachen.de (T.F.E.);; 2Center for Integrated Oncology (CIO), University Hospital RWTH Aachen, 52074 Aachen, Germanysleypold@ukaachen.de (S.L.); nadine.gaisa@uniklinik-ulm.de (N.T.G.); 3Department of Urology and Kidney Transplantation, Martin Luther University, 06097 Halle (Saale), Germany; 4Interdisciplinary Center for Clinical Research (IZKF), RWTH Aachen University, 52074 Aachen, Germany; 5Institute of Pathology, University Hospital RWTH Aachen, 52074 Aachen, Germany; 6Institute of Pathology, University Hospital Ulm, 89081 Ulm, Germany

**Keywords:** prostate cancer, targeted drug delivery, diagnostics, folic acid receptor, glycosylphosphatidylinositol transamidase

## Abstract

**Simple Summary:**

Folate receptors can serve to increase folate uptake in cancer cells through significant overexpression, which is ultimately critical for cell proliferation and tumor growth. Therefore, the characterization of folate receptor expression in cancer presents exciting opportunities in multiple dimensions. On the one hand, the folate receptor can be used as a target for selective transfer of therapeutic agents. On the other hand, it can be used for risk stratification. We found that in prostate cancer (PCa), activation and membrane integration by GPI–transamidase exhibit significant changes. By image analysis, we could demonstrate these differences to healthy control cells and tissues, and we expect high potential in this approach to identify biological subtypes and allow risk stratification based on these alterations in PCa.

**Abstract:**

Due to the proliferation-induced high demand of cancer cells for folic acid (FA), significant overexpression of folate receptors 1 (FR1) is detected in most cancers. To our knowledge, a detailed characterization of FR1 expression and regulation regarding therapeutic and diagnostic feasibilities in prostate cancer (PCa) has not been described. In the present study, cell cultures, as well as tissue sections, were analyzed using Western blot, qRT-PCR and immunofluorescence. In addition, we utilized FA-functionalized lipoplexes to characterize the potential of FR1-targeted delivery into PCa cells. Interestingly, we detected a high level of FR1-mRNA in healthy prostate epithelial cells and healthy prostate tissue. However, we were able to show that PCa cells in vitro and PCa tissue showed a massively enhanced FR1 membrane localization where the receptor can finally gain its function. We were able to link these changes to the overexpression of GPI–transamidase (GPI-T) by image analysis. PCa cells in vitro and PCa tissue show the strongest overexpression of GPI-T and thereby induce FR1 membrane localization. Finally, we utilized FA-functionalized lipoplexes to selectively transfer pDNA into PCa cells and demonstrate the therapeutic potential of FR1. Thus, FR1 represents a very promising candidate for targeted therapeutic transfer pathways in PCa and in combination with GPI-T, may provide predictive imaging in addition to established diagnostics.

## 1. Introduction

Prostate cancer (PCa) is the second most common cancer in men, with >1.4 million cases reported in 2021, and is the fifth leading cause of cancer-related death, with >375.000 deaths worldwide [1]. While validated diagnostic methods are available, including prostate-specific antigen (PSA) diagnostics and continuously optimized biopsies, including modern imaging techniques [2,3], there is still a lack of early markers that can differentiate initial stages of the disease, such as high-grade prostatic intraepithelial neoplasia (HG-PIN) [4], and that can also be used for more targeted therapeutic predictions and interventions [5]. Here, research is mainly focused on modern approaches that aim to enable liquid biopsies by using, e.g., miRNAs [6]. Especially, therapies for metastatic castration-resistant PCa continued to evolve. Besides androgen deprivation therapy (ADT) and chemotherapy, combination therapies consisting of various ADTs became especially more prevalent [7]. Innovative therapeutic concepts, such as gene therapies, are currently in clinical trials and could significantly improve the treatment of these patients [8]. For example, functionalized liposomes could be used for targeted transfer of these innovative therapeutics. In this context, mainly receptors that induce active endocytosis of cells are used; as such, receptor-mediated endocytosis can achieve significant improvements in the targeted transfer of therapeutics. For this purpose, folate receptors (FR) have already been successfully utilized in numerous applications [9,10,11]. Moreover, PEGylation can be used to reduce transfer efficiency, the so-called PEG dilemma, which can ultimately be used in combination with functionalization of lipoplexes such as folic acid (FA) to compensate PEG-effects and allow selective enhancement of nucleic acid transfer into cancer cells [12]. In other cancer types, such as breast cancer, selective uptake of therapeutics via specific receptors has been successfully achieved, and clinical success has been reported [13]. Moreover, in both healthy prostate and prostate cancer, folate metabolism has a substantial metabolic function. Overexpression of prostate-specific membrane antigen (PSMA), a folate hydrolase that allows the uptake of alternative folate sources such as polyglutamated FA, also demonstrates its importance in prostate cancer growth [14]. However, FR also plays a significant role in healthy prostate tissue, as the prostate is massively dependent on folate one-carbon metabolism for the synthesis of polyamines and polyamine spermine. Therefore, detailed characterization, especially in relation to its activating moiety glycosylphosphatidylinositol transamidase (GPI-T), is of particular importance for the evaluation of potential diagnostic and therapeutic applications of FR in PCa [15,16]. Thus, for example, Patil et al. treated PSMA-positive cells with FA-functionalized lipoplexes and demonstrated significant enhancement and selective uptake in cancer cells [17]. Alserihi et al. also pursued this strategy and demonstrated a corresponding improvement in the anticancer activity of epigallocatechin 3-gallate (EGCG) in cell culture [18]. However, all these strategies have not yet induced a major therapeutic impact, partly as a consequence of the missing characterization of FR alterations in prostate cancer. The overall expression of FR in prostate cancer has been described [19], indicating a corresponding increase in its expression with increasing cancer severity. In terms of diagnosis, accurate characterization of primary tumor material has not yet been performed. However, Lian et al. describe a possible prognostic correlation between FR-positive circulating tumor cells and the corresponding tumor situation of patients despite low PSA levels in early tumor stages [20]. In addition to the investigation of FR-expression and localization, GPI-T is also a promising prognostic candidate that is responsible for the ultimate activation and membrane localization of FR in cancer cells [21]. This protein shows increased expression in a variety of tumors. In prostate cancer, this increased expression was already detected in 41% of the cancer tissues. While more significant changes were detected than in lung carcinoma [22], no prognostic correlation was examined here. We specifically focused on the simultaneous characterization of GPI-T and FR to understand the correlations and alterations in PCa and thus establish prognostic as well as therapeutic recommendations based on these molecular markers.

## 2. Materials and Methods

### 2.1. Cell Culture

For all experiments, the two cell lines, LNCaP (Merck, 89110211, prostate cancer human, Rahway, NJ, USA) and PNT2 (Merck, 95012613, healthy epithelial prostate cell), were used. A third cell line, VCaP (Merck, 06020201, prostate cancer human), was used for further validation of the results. All cells were cultivated at 37 °C and saturated humidity with 5% CO_2_. For cell cultures of LNCaP and VCaP, all substrates were coated with fibronectin (FN) (Merck, F0635) prior to cell seeding. For better comparability, FN-coating was also performed for PNT2 cells prior to transfections. FN was dissolved in demineralized water to a concentration of 1 g/L and then further diluted in phosphate-buffered saline (PBS) to a final concentration of 10 mg/L and incubated on the substrates at 37 °C for 30 min.

LNCaP and PNT2 were cultured in RPMI-1640, supplemented with 10% (*v*/*v*) Fetal Bovine Serum Superior (FBS Superior) (Merck, S0615) and 1% (*v*/*v*) penicillin–streptomycin (10,000 U/mL) (Thermo Scientific, 15140122, Waltham, MA, USA). For PNT2 cell culture, 2 mM glutamine (Thermo Scientific, A2916801) was additionally added to the RPMI-1640 medium. VCaP were cultivated in DMEM (Thermo Scientific, 11965092), supplemented with 10% (*v*/*v*) Fetal Bovine Serum Superior (FBS Superior) (Merck, S0615) and 1% (*v*/*v*) penicillin–streptomycin (10,000 U/mL) (Thermo Scientific, 15140122). The medium was changed every 2–4 days during the cell growth. All cells were subcultured at 80–90% cell density and transferred to fresh T75 flasks. Cell detachment was performed with 0.25% trypsin–EDTA solution (Thermo Scientific, 25200056) for 3 to 10 min. A hemocytometer was used for cell counting.

### 2.2. Tissue Sections

A total of 9 formalin-fixed and paraffin-embedded (FFPE) human prostate samples that originated from the years 2021 to 2023 were used for tissue characterizations. In total, three healthy control samples, removed during prostate enucleation for benign prostatic hyperplasia, and six confirmed carcinoma biopsy samples, resected by robotic-assisted prostatectomy, were used. The use of patient samples was approved by the local ethics committee of the medical faculty of the Rheinisch Westfälische Technische Hochschule (RWTH) Aachen with the internal reference EK23-043.

### 2.3. Quantification of RNA Transcripts

#### 2.3.1. RNA Isolation

For RNA isolation from cell cultures, the RNeasy-Midi Kit (QIAGEN, 77144, Silvercord, Hongkong, China) was used. Cells were cultivated in T75 flasks and subsequently processed according to the manufacturer’s instructions. The cells were seeded 3 days prior to processing at a density of 40,000 cells/cm^2^ for LNCaP, 26,666 cells/cm^2^ for PNT2 and 133,333 cells/cm^2^ for VCaP.

RNA was dissolved in 300 µL nuclease-free water (Promega, MC1191, Madison, WI, USA). The concentration was determined using a NanoDrop (Thermo Scientific, A38189), and RNA was stored at −150 °C until use.

For RNA Isolation from prostate tissue, the ReliaPrep™ FFPE Total RNA Miniprep System (Promega, Z1001) was used. Depending on the size of the samples, FFPE tissue sections with a total of up to 2 mm^3^ were solved in mineral oil and processed according to the manufacturer’s instructions. Each RNA isolate is eluted in 40 µL nuclease-free water (Promega, MC1191), and its concentration determined using NanoDrop (Thermo Scientific, A38189).

#### 2.3.2. cDNA Synthesis

For cDNA synthesis of isolated RNA (Section 2.3.1), the RNA solution was thawed, further steps were accomplished on ice, and the Maxima First Strand cDNA Synthesis Kit (Thermo Scientific, EP0751, EP0741) was used. For each cell type, 300 ng RNA was diluted, and for each tissue sample, 500 ng RNA was diluted with nuclease-free water (Promega, MC1191) to a final volume of 14 µL. Finally, 4 µL 5x Reaction Mix and 2 µL Maxima Enzyme Mix were added.

The cDNA synthesis was performed by incubation at 25 °C for 10 min, 50 °C for 30 min, 85 °C for 5 min and finally, cooling down to 4 °C in a thermocycler (analytik-jena, Biometra TRIO).

#### 2.3.3. qRT-PCR

For qRT-PCR, TaqMan probes were used, which were directed against the GPI-T subunit PIGK (Thermo Scientific, Hs00300778_m1), FR1 (Thermo Scientific, Hs06631528_s1) and glyceraldehyde 3-phosphate dehydrogenase (GAPDH) (Thermo Scientific, Hs02786624_g1). An amount of 1 µL Expression Assay, 10 µL TagMan Expression Mastermix and 6 µL nuclease-free water was added to 3 µL cDNA (see Section 2.3.2) in a 96-well plate. DNA replication was performed and quantified using the StepOnePlus Real-Time PCR System (Thermo Scientific, 4376600). For this purpose, the test mixtures were heated to 95 °C for 20 s, followed by 40 cycles of cooling to 60 °C for 20 s and heating up to 95 °C for 1 s.

### 2.4. Quantification of Proteins

#### 2.4.1. Protein Isolation

All preparation steps were performed on ice. Cells were seeded 3 days prior to isolation in T75 flasks. Cells were detached into ice-cold PBS using a cell scraper. After centrifugation at 100 rcf and 4 °C for 5 min, the supernatant was discarded. For cell lysis, cells were incubated in 150 µL RIPA lysis buffer plus inhibitors (see Table 1) and homogenized with a syringe and 0.26 mm inner diameter needle. The isolate was incubated on ice for 30 min and then centrifuged at 8000 rcf and 4 °C for 15 min. The supernatant was stored at −80 °C until use.

The Qproteome FFPE Tissue Kit (Qiagen, 37623) was used for protein isolation from tissue samples. Ten-micrometer-thick FFPE sections were used; the number of sections was based on the size of the tissue areas. The sections were deparaffinized using xylol and ethanol, transferred to the EXB Plus extraction buffer and incubated at 100 °C for 20 min. After 2 more hours at 80 °C and an agitation of 750 rpm, the containers were centrifuged for 15 min at 4 °C and 14,000× *g*.

#### 2.4.2. Determination of Protein Concentration

The Micro BCA Protein Assay Kit (Thermo Scientific, 23235) was used to determine the protein concentration of the obtained isolates (see Section 2.4.1). A dilution series of BSA in RIPA lysis buffer was prepared as described by the manufacturer. This dilution series and the samples (previously diluted 1:30 in RIPA lysis buffer) were mixed with the Working Reagent (WR), which shifts its absorption maximum after binding to proteins. Subsequently, the absorption at 562 nm was measured using the SpectraMax iD3 (MolecularDevices, San Jose, CA, USA). The protein concentrations of protein isolates were determined using the dilution series-standard curve.

#### 2.4.3. SDS Page

Prior to gel electrophoresis, protein isolates (see Section 2.4.1) were diluted with RIPA lysis buffer to identical protein concentrations. Each sample was loaded onto a TGX stain-free protein gel (Bio-Rad, 4568026, Hercules, CA, USA) in a total volume of 15 µL. Sample dilution (3 sample/1 loading dye) was performed with loading dye (Bio-Rad, 1610747), previously supplemented with β-mercaptoethanol (Merck, 8.05740.0250) (1 β-mercaptoethanol/9 loading dye). As protein standard, the gel was loaded with Precision Plus Protein All Blue Prestained Protein Standards (Bio-Rad, 1610373). To separate proteins in TGX stain-free protein gels, a voltage of 120 V was applied for 2 h.

#### 2.4.4. Antibodies

All primary and secondary antibodies used are listed in Table 2, indicating each specific dilution depending on specific experiments.

#### 2.4.5. Western Blot

An LF-PVDF membrane and transfer stacks (Bio-Rad, 1704274) were used for blotting and prepared according to the manufacturer’s specifications. The blot was placed in the Trans-Blot Turbo Transfer System (Bio-Rad, 1704150) for 7 min at 1.3 A and 25 V using the program Mixed MW (Turbo).

By using a blocking buffer containing 5% (*w*/*w*) BSA (Merck, A-7906) and 0.05% (*v*/*v*) Tween 20 (Merck, P1379) in PBS at 4 °C overnight, all unspecific binding sites were saturated.

Incubation of primary antibodies (see Table 2) was performed in a blocking buffer overnight at 4 °C (FR1), respectively, at room temperature (RT) for one hour (GPI-T). After washing three times with blocking buffer for 5 min, secondary antibodies (see Table 2) were added to the membrane in blocking buffer and incubated for 1 h at RT. After washing in PBS three times, the membrane was imaged at emission wavelengths of 520 nm and 720 nm using the ChemiDoc MP Imaging System (Bio-Rad, 12003154). By using the stain-free protein gels, a loading control could be implemented on the total amount of protein effectively transferred to the PVDF membrane. A corresponding quantification and calculation of the detected proteins (FR1/GPI-T) was carried out with the ImageLab software (Version 6.1) using additional background quantifications. The original, uncropped Western blot membrane can be found in Appendix A.

#### 2.4.6. Immunofluorescence Staining

##### Cell Culture

Immunofluorescence staining of cell cultures was performed on glass substrates (µ-dish, ibidi, 81218-200) seeded 24 h prior to staining. In brief, cells were fixed with 4% formaldehyde (Otto Fischar, 02653048, Saarbrücken, Germany) for 30 min at 37 °C, followed by partial membrane permeabilization in 2.5% (*v*/*v*) Triton X-100 (Merck, X100) in PBS for 3 min, allowing a membrane protein-targeting immunofluorescence. This was followed by the blocking of non-specific binding sites for 45 min in a blocking buffer consisting of 5% (*w*/*w*) BSA in PBS.

Primary antibodies (see Table 2) were incubated in a blocking buffer for 1 h. After washing in PBS three times for 5 min, secondary antibodies (see Table 2) were added to the blocking buffer and incubated for 1 h. Finally, nuclei staining was performed by incubation in 1:20,000 diluted Hoechst 33342 Ready Flow Reagent (Thermo Scientific, R37165) in PBS for 5 min, followed by washing in PBS for 5 min. By sealing treated cells with a coverslip (Fisher Scientific, 15767572, Hampton County, NH, USA) over Immu-Mount (Epredia, 9990402, Kalamazoo, MI, USA), samples were preserved for subsequent microscopy.

##### Tissue

For immunofluorescence, 3 µm FFPE tissue sections of prostate cancer and benign tissue were used. For deparaffinization, these slides were incubated sequentially for 5 min in ROTI Histol (Carl Roth, 6640.1), descending ethanol solutions (100%, 95% and 70%), and finally demineralized water.

Antigen retrieval was then performed by incubation of slides in 1:100 diluted antigen unmasking solution (VectorLaboratories, H-3300-250, Newark, CA, USA) in demineralized water. Tissue sections and unmasking solution were then heated in a microwave at 900 W until boiling. Once boiling was reached, the solution was heated for an additional 10 min at 180 W. The solution was then cooled to RT for 30 min, followed by partial membrane permeabilization by incubation for 5 min in 0.05% (*v*/*v*) Tween 20 in PBS. Then, tissue slices were encircled using a hydrophobic DAKO pen (Agilent, S2002, Beijing, China) to ensure the remaining liquid was subsequently added to the tissue. After that, non-specific binding sites were blocked by incubation in a blocking buffer consisting of 5% (*w*/*w*) BSA for 1 h at RT in a humid staining chamber.

Incubation of primary antibodies (see Table 2) in blocking buffer was carried out overnight at 4 °C in a humid staining chamber. The next day, secondary antibodies (see Table 2) were added to the blocking buffer and incubated in a humid staining chamber at RT for 1 h. Subsequently, cell nuclei were fluorescently labeled by incubation in 1:20,000 diluted Hoechst 33342 Ready Flow Reagent (Thermo Scientific, R37165) in PBS for 5 min, followed by washing in PBS for 5 min. Finally, the sections were covered with 2 drops of Immuno-Mount and a coverslip (Engelbrecht, K12460, Edermünden, Germany).

### 2.5. Transfection

#### 2.5.1. Preparation of Standard Lipoplexes—Lipofectamine 3000

To transfer eGFP-plasmid (PlasmidFactory, PF464, Bielefeld, Germany), cells were seeded 24 h prior to transfection in 24-well plates coated with fibronectin as described in Section 1. Lipofectamine 3000 was prepared as described previously by Hoffmann et al. [12] to ensure comparable lipoplex characteristics compared to functionalized lipoplexes (Section 2.5.2). In brief, 2.5 μg plasmid, 2.5 μL P3000 and 5 μL Lipofectamine 3000 were incubated, and 4 μL of this solution was added to each well of the respective cell cultures to transfer 1 µg plasmid-DNA. LNCaP and PNT2 were analyzed 24 h after transfection, and VCaP cells were analyzed 72/120 h after transfection supplemented with an additional 2 mL fresh medium 24 h after transfection.

#### 2.5.2. Functionalization of Lipoplexes

To functionalize lipoplexes, folate–PEG–NHS (PEG-FA) (Nanosoft Polymers, SKU: 11395, New Taipei City, Taiwan) solutions were prepared in demineralized water in concentrations of 20 mM, 2 mM and 0.2 mM. An amount of 1 µL of each solution was then added to 4 µL of standard lipoplexes (Section 2.5.1) to dilute them further to concentrations of 4 mM, 0.4 mM and 0.04 mM. After an incubation period of 5 min, functionalized lipoplexes were added to each well of the cell culture plates. Quantification was performed 24 h after transfection by fluorescence microscopy and flow cytometric analysis.

For flow cytometric samples, cells were separated by 0.25% Trypsin–EDTA, as described in Section 2.1. Cells were resuspended in 200 µL PBS, and the suspension was measured as described in Section 2.7.

#### 2.5.3. Transfection and Immunofluorescence Staining

To characterize the selectivity and transfer efficiency of eGFP-plasmid transfections (Section 2.5.1 and Section 2.5.2) in co-cultures, immunofluorescence staining of PNT2 and LNCaP cells was performed. For this purpose, 24-well plates that were coated with fibronectin, as described in Section 1, were seeded with 25,000 cells/cm^2^ of each cell type. In addition to co-cultures, pure cultures of both cell types were seeded for subsequent gating in flow cytometric analysis.

Immunofluorescence staining of cells was performed in suspension. Cells were separated by incubating in trypsin–EDTA (0.25%), as described in Section 1. Subsequently, cells were fixed with 1% formaldehyde, using 4% formaldehyde diluted in PBS, at 37 °C for 1 h. Afterward, blocking of non-specific binding sites by incubation in blocking buffer (PBS containing 5% [*w*/*w*] BSA) for 30 min was performed, followed by incubation with the primary antibody (see Table 2) in blocking buffer for 1 h. After that, the secondary antibody (see Table 2) was added to the blocking buffer at a 1:400 dilution and incubated for 1 h. Cells were transferred to 5 mL tubes (Corning, 352054, Corning, NY, USA) in 200 µL PBS for subsequent flow cytometric analysis (Section 2.7.2).

### 2.6. Microscopy

A Leica DM IL LED microscope (Leica-Microsystems, Wetzlar, Germnay) was used for microscopy of cell culture substrates. As a light source, pE-300lite (CoolLED) was applied with an electromagnetic spectrum that peaks at around 400 nm, 450 nm and 580 nm. The filters used were Y3 ET (Leica-Microsystems, 11525311), DM1000 (Leica-Microsystems, 11513824) and GFP ET (Leica-Microsystems, 11504174). For the characterization of eGFP-positive cells, HI PLAN 4×/0.10 was used; for the characterization of co-cultures after transfection and immunofluorescence, HI PLAN CY 10×/0.25 was used; and for immunofluorescence staining of cell culture, HI PLAN I 40×/0.50 or HI PLAN I 20×/0.30 was used.

The K3M camera (Leica-Microsystems), integrated into the microscope, and the corresponding LAS X-software (Version 3.7.4.23463) was used for image acquisition with a resolution of 3072 × 2048 pixels. The settings (exposure time and gain) were selected depending on the substrate and kept constant within a series of experiments, as described in Table 3.

The Zeiss Axio Observer microscope with the Axiocam 503 mono was used for tissue immunofluorescence with a Colibri 5 as light source and excitation filters 370–410 nm (DAPI), 450–490 nm (GPI-T) and 538–562 nm (FR1), as well as emission filters of 430–470 nm (DAPI), 500–550 nm (GPI-T) and 570–640 nm (FR1).

Each image was recorded in four channels. Due to deviating focal planes in the samples, the channels for EGFP and FR, as well as phase contrast and DAPI, were recorded in the same plane and subsequently merged into one image file.

### 2.7. Flow Cytometry

#### 2.7.1. FACS Canto II

The FACS Canto II (BD Biosciences, Franklin Lakes, NJ, USA) system with 3-laser (405 nm, 488 nm, 633 nm) and a configuration for 8 fluorescent parameters (2-4-2) was used for flow cytometric analysis of the transfected cells from Section 2.5.1 and Section 2.5.2. Initially, cells were gated by size and granularity using forward scatter (FSC) and side scatter (SSC) to differentiate between cells and cell debris. The B1 channel of the blue laser (488 nm) with a 530/30 bandpass filter and a 502 longpass filter allowed for detecting transfected cells via the reporter gene eGFP. 

#### 2.7.2. LSRFortessa

For the combination of eGFP- and GPI-T-immunofluorescence characterization (Section 2.5.3), LSRFortessa (BD Biosciences) was used, a 4-laser system (405 nm, 488 nm, 561 nm, 640 nm) with a configuration for 16 fluorescent parameters (6-2-5-3). Cells were gated by FSC and SSC before eGFP-positive cells were detected by the blue laser (488 nm) with the 530/30 bandpass filter. Furthermore, the yellow/green laser (561 nm) and the 670/30 filter were used to detect the fluorophore Alexa Fluor 594, which was used to stain GPI-T expression in cancer and healthy cells, as described in Section 2.5.3.

### 2.8. Image Analysis

Image-processing of immunofluorescence images was performed using “ImageJ” software (Version v1.54f). To analyze protein expression at single-cell level, single cells were randomly selected in DAPI channels, and phase-contrast images were used to define polygons around single cells, based on the geometry of the selected cells, with the “Polygon selections” tool. The “RecordD.” tool was used to track selected cells, allowing the coordinates of the drawn lines and polygons to be determined and saved as a text file. Subsequently, the fluorescence intensity of the surface of these selected polygons and their area were measured using the “Measure” function in the respective fluorescence channel.

For the characterization of fluorescence profiles, cells were selected and labeled as described above. Subsequently, lines were drawn through the center points of cells across the entire cell using the “ROI Manager” tool with a macro that allowed for creating a set of 180 equidistant lines, intersecting each other in the center of the cell and subsequently creating a circle-like shape. Fluorescence intensities were examined along these lines. The “Multi Plot” function was used for this purpose, followed by data output using the “List” button. Fluorescence profiles were created for each cell by taking average values of each of the measured points of all 180 lines. In addition, fluorescence profiles were created from these mean values by further averaging all cells of an investigated sample, showing the fluorescence intensity along cell geometries of that population. 

To characterize the correlation of GPI-T fluorescence intensity and FR1 fluorescence in the cell membrane, individual cells were characterized based on both fluorescence channels as described above. The results of this characterization were then combined in a scatter plot with 10 cells each, individually analyzed and merged into one data point.

### 2.9. Statistical Analysis

All data are given as mean, including standard deviation of at least three independent measurements. Relative calculated data were generated from each experiment and combined accordingly from the different individual experiments to give a relative mean and the corresponding standard deviation. Statistical analysis (univariate ANOVA) was performed with Microsoft Excel (MS Office 2019) for multiple comparisons. Figures and graphs were generated using Origin 2019 64Bit (OriginLab Graphing and Analysis). 

Only significant differences between individual probes were marked in graphs with asterisks. For this purpose, a *p*-value of ≤0.05 was considered as significant. *p*-values of 0.05, 0.01 and 0.001 were labeled with one to three asterisks.

## 3. Results

Folate receptor 1 (FR1) expression was characterized in healthy PNT2 prostate epithelial cells and LNCaP prostate cancer cells. Surprisingly, high levels of FR1-mRNA were found in healthy prostate epithelial cells (Figure 1A). However, quantification of cellular FR1-protein levels by Western blot found no significant difference (Figure 1B) between healthy and prostate cancer cell lines. Using a membrane protein-targeting immunofluorescence, FR1-signal intensity confirmed a significant overexpression in LNCaP cells (Figure 1C,D), indicating an enhanced activation and membrane localization of FR1 in malignant prostate cells. To validate these differences, occludin staining was also carried out to detect the basic staining of cellular plasma-membranes of both cell types. This revealed comparable occludin staining intensities in PNT2 and LNCaP, which allowed the specificity and differences in FR1 signal intensity to be validated (Appendix A).

Consequently, the expression profile of the FR1-activating complex glycosylphosphatidylinositol transamidase (GPI-T) was characterized in PNT2 and LNCaP cells. This showed a distinct and significant pattern in all screening methods with approximately 4-fold increased mRNA levels (Figure 2A), up to 4-fold increased protein levels after Western blot assays (Figure 2B), and 2-fold increased signal intensities of GPI-T by immunofluorescence (Figure 2C) in LNCaP cells. In addition, co-staining of FR1 and GPI-T (Figure 2D) further indicated the difference in FR1 localization between PNT2 and LNCaP. The more distinct membrane signals of FR1 in LNCaP cells are correlated with respective higher GPI-T levels. This correlation could be illustrated by performing image analysis on the FR1-localization and GPI-T signal intensity (Figure 2E). GPI-T-based activation and resulting localization of the receptor into the cellular plasma membrane could ultimately induce the crucial functionality of FR1 and its potential use as a target receptor for the transfer of therapeutics into cancer cells. 

To characterize the potential of FR1 as a target receptor for the selective transfer of therapeutics, standard and FA-functionalized lipoplexes were used to transfer eGFP-pDNA in healthy PNT2 and LNCaP cancer cells. Standard lipoplexes without FA-functionalization were used to transfer eGFP-pDNA as a control in both cell types (PNT2 and LNCaP). Additionally, the same lipoplexes were functionalized by NHS chemistry using different amounts of PEG-FA (0.04–4 mM). The transfection efficiencies were examined by fluorescence microscopy (Figure 3A,B) and quantified by flow cytometry (Figure 3C). Alterations in transfer efficiencies for each cell type clearly show that a more accelerated decrease in transfer efficiency is induced in healthy PNT2 cells by the PEG dilemma, and FA functionalization can more distinctively optimize transfer efficiency in LNCaP cells. For PNT2, a significant reduction of more than 30% was already detectable at 0.4 mM PEG-FA. At this concentration, LNCaP cells showed a non-significant but slight increase compared to non-functionalized eGFP-lipoplexes (Figure 3C). In addition, flow cytometry assays were used to characterize the differential specificity of DNA transfer using FA-functionalized lipoplexes. Here, improvements in the specificity of 9–32% were shown for the corresponding PEG-FA functionalized lipoplexes with significant improvements for 0.04 mM (32% s.d. 35%) and 4 mM (9% s.d. 11%) (Figure 3D). Comparable results were obtained for the VCap prostate cancer cell line, where a slight, non-significant improvement was obtained with 0.4 mM PEG-FA, resulting in a significant improvement for selectivity of 112% (s.d. 157%; see Appendix A). To validate this specificity and selectivity, a co-culture of PNT2 and LNCaP cells was prepared, and standard lipoplexes were characterized in comparison to FA-functionalized lipoplexes. In addition, GPI-T-immunofluorescence was used to differentiate between healthy PNT2 and LNCaP cancer cells. It was shown that GPI-T-immunofluorescence allowed a clear separation of both cell types due to lower signal intensities of GPI-T-immunofluorescence in PNT2 cells (white boxes, Figure 3E,F). Moreover, fluorescence microscopy images indicated an improved selectivity of eGFP-plasmid DNA transfer by FA-functionalized lipoplexes. To further quantify this improved selectivity, equally processed cells (transfection of standard lipoplex or functionalized lipoplex followed by GPI-T-immunofluorescence) were analyzed by flow cytometry. A significant and more than 20% enhanced eGFP-pDNA transfer by FA-functionalization into the accordingly more intense GPI-T-stained LNCaP cells could be detected (Figure 3G). 

To verify the clinical significance of these findings, identical FR1 and GPI-T characterizations were performed on healthy prostate specimens (*n* = 3) and PCa sections (*n* = 6). Here, qRT-PCR-analysis (Figure 4A) of GPI-T- and FR1-mRNA quantification confirmed cell culture results with significantly increased FR1-mRNA values for the healthy sections (1.0 s.d. 0.36) compared to PCa patients (0.34 s.d. 0.19). For GPI-T, non-significantly increased mRNA values for PCa patients (1.2 s.d. 0.59) were detected compared to healthy control sections (1.0 s.d. 0.29). Likewise, protein quantification (Figure 4B) showed non-significant changes in PCa sections with 1.14-fold (s.d. 0.32) increased values of FR1 and 1.37-fold (s.d. 1.73) increased GPI-T-protein amount compared to healthy tissue sections. Even the quantification of single-cell fluorescence intensities in immunofluorescence images (Figure 4C,D) with FR1 and GPI-T did not allow a precise differentiation between healthy and PCa sections (see Appendix A). However, altered localizations of FR1-signals could also be detected in single-cell analysis of tissues from PCa sections compared to healthy tissue. Linear cell profiles of healthy (Figure 4E) and PCa tissue (Figure 4F) indicated a significant signal shift in cancer cells with 1.31-fold (s.d. 0.6) increased fluorescence intensity in the cell membrane compared to FR1-membrane signal intensities of healthy cells (Figure 4G). Moreover, a clear correlation (Pearson R = 0.801) between GPI-T expression and resulting FR1-membrane localization was observed in tissue experiments. (Figure 4H).

## 4. Discussion

Based on the present data, we were able to show for PCa that despite high FR1-mRNA expressions in healthy prostate cells (cell culture and tissue studies), this receptor can be utilized for selective transfer of therapeutics (here pDNA). Additionally, the regulation and activation of FR1, with subsequent localization in cell membranes, could be used by cellular localization studies to differentiate healthy and PCa tissue sections with high sensitivity. These results demonstrate for the first time the potential of this receptor in PCa for diagnostic purposes using the image analyses established here. In the following sections, these results are related to existing studies from PCa and other types of neoplasms.

### 4.1. Expression Pattern of Folate Receptor in Prostate Cancer

Over-expression of FR1 has already been described in various tumors such as ovarian cancer [23,24], lung cancer [25] and glioblastoma [26]. The enhanced receptor localization with approximately 4-fold higher immunofluorescence intensities compared to healthy cells is lower than th overexpressions described for breast cancer and glioblastoma with an 18-fold increase compared to healthy control cells [12]. PNT-2 cells and healthy tissue likewise exhibited higher levels of FR1-mRNA compared to LNCaP-cells and cancer tissue. On the one hand, prostate cells use a C1 metabolic pathway with folates as a substrate for polyamine synthesis, especially spermine [15]. Therefore, healthy prostate cells also require high amounts of folates in vivo and could realize a correspondingly rapid adaptation of folate uptake for spermine synthesis by comparatively high FR1-mRNA amounts. On the other hand, FR1 is known to function as a transcription factor and binds cis-regulatory elements at promoter regions of Fgfr4 and Hes1 [27]. To our knowledge, however, a function as a transcription factor in the prostate has not been described so far, but according to these findings, it could also explain an increased mRNA expression.

The increased FR1 expression level indicated by immunofluorescence can certainly be explained by the staining method used, targeting membrane-bound proteins like FR1 by reduced detergent incubations [28] compared to standard immunofluorescence protocols [29,30]. Therefore, a strong FR1 signal can be detected, particularly in tumor cell membranes, where, consequently, a large amount of active FR1 is localized. Western blot analysis, as shown here, further confirms the effect of specific regulation and processing of FR1 in healthy PNT2 and LNCaP-PCa cells, as similar FR1 protein levels lead to different levels of activated FR1 proteins on cell membranes (Figure 2) and active transport of FA-associated lipoplexes (Figure 3). 

### 4.2. Regulation of Folate Receptors in Prostate Cancer

Several mechanisms have an impact on the transcription and translation of mRNAs. One of those mechanisms is RNA interference by miRNAs. Regulation of miRNAs is described as a prominent mechanism of prostate cancer progression [31,32]. For example, microRNA-29b, which is downregulated in prostate cancer [33,34], negatively regulates FR1-mRNA [35]. Besides FR1, prostate cancer cells show a specific overexpression of PSMA, a folate-associated membrane protein that functions as a folate hydrolase [36]. PSMA provides an alternative folate source for prostate cancer cells, highlighting the importance of FA for this disease. Regarding folate metabolism, overexpression of PSMA causes a proliferative advantage, as additional folates can be synthesized by PSMA from their polyglutamylated form [14].

FR1 is a membrane-bound protein anchored to the cytoplasmic membrane by a GPI anchor [37]. Therefore, GPI-T conditions the activation of FR1 in the context of recruitment to the cytoplasmic membrane, and consequently, high levels of GPI-T argue for enhanced uptake of folates through increased activation and membrane localization of FR1. The localization of FR1 was characterized by immunofluorescence in this work. However, a verification of FR1-localization, for instance, by Förster resonance energy transfer (FRET), could be used to enable additional insights into the receptor’s localization [38,39]. 

Different subunits of GPI-T are differently expressed in cancers with specific up- and downregulation. For bladder cancer, an upregulation of the PIG-U-subunit and downregulation of the PIG-K-subunit are described [21], while for gastric cancer, an upregulation of GPAA1 was described [40]. For prostate cancer, an increase in the PIG-K subunit was also described by Nagpal et al. (2008) [21], which, however, showed smaller alterations compared to results in other types of cancer. The potential use of these GPI-T subunits as a prognostic tool is already being discussed [40,41,42]. In hepatocellular carcinoma (HCC) patients, PIG-U was successfully tested for this purpose [41]. In this regard, PIG-U is suspected to be involved in cell cycle regulation in HCC patients, and consequently, overexpression of this protein indicated a poor survival prognosis for patients [41]. Breast cancer cells have shown an overexpression of PIG-U and PIG-T [42]. This likely leads to an increased number of GPI-anchored proteins and contributes to tumor invasiveness and metastasis [42]. GPAA1, as another subunit of GPI-T, was shown to be highly upregulated in gastric cancer. Overexpression of this gene was shown to contribute to cancer growth and metastasis [40]. As various GPI-T-subunits are being overexpressed in different tumors, including prostate cancer, the here quantified PIG-K-subunit, in particular, could also prove to be a promising prognostic tool. However, the characterization of further subunits could provide additional prognostic indications for prostate cancer-specific expression patterns.

The correlation of FR1 localization and GPI-T expression found here particularly indicates substantial alterations of FA uptake and cellular characteristics and may influence diagnostics as well as potential use for targeted drug delivery. Previously, Lian et al. demonstrated that circulating tumor cells showing FR overexpression may imply a poor prognosis for prostate cancer patients [20]. We have not yet been able to explicitly link the exact correlation of the results described here to clinical courses, but we are planning more in-depth studies that might enable predicting the risk of metastasis or progression by this type of staining. 

### 4.3. Clinical Utility of Prostate Cancer Folate Receptor Activation

While an extended analysis and correlation of these localization and expression changes to clinical parameters and risk stratifications is still pending, our data illustrate that only a correlation of GPI-T and FR1, as well as a localization study, enables healthy tissues to be distinguished from tumors on cellular levels. Further patient-specific investigation with corresponding clinical parameters could provide information about the diagnostic potential of these proteins and possibly enable new possibilities for the differentiation of a biological subset of tumor types. The altered activation and membrane localization of FR1, being highly upregulated in PCa cells, also allows for linking targeted therapy to the expression of GPI-T and FR1. As a model for the therapeutic use of FR1 in prostate cancer cells, we used a PEG-FA functionalized lipofection system, previously described by Hoffmann et al. in 2022 [12]. Functionalization of endocytic lipoplexes with FA in combination with PEGylation allows for investigating FR1-driven and selective uptake of pDNA. Thus, we demonstrated that, despite similar protein levels of FR1, selective and specific uptake of functionalized lipoplexes can occur in prostate cancer cells as a result of the previously described activation of FR1 by GPI-T. This demonstrated impressively that next to its diagnostic potential, FR1 may provide a therapeutical target to selectively internalize drugs into prostate cancer cells by functionalized lipoplex formulations. Its clinical use as a target receptor has already been shown in other FR1-positive solid carcinomas, such as ovarian cancer, endometrial cancer and breast cancer [43]. For example, MORAb-202, a monoclonal antibody conjugate against FR1, induces apoptosis [44] and transports the conjugated eribulin mesylate into cancer cells, which is an approved drug for the treatment of metastatic breast cancer and soft tissue sarcoma by inhibiting microtubule growth [45].

### 4.4. Limitations and Clinical Significance

The ubiquitous distribution of FR1 in almost all tissues makes a specific investigation of its activation and localization essential. At this point, even in previous successful clinical applications, only an efficient pre-selection of a therapeutic agent, e.g., MORAb-202, could be realized by means of FA [25,43]. These forms of a therapeutic agent always require a second level of selection, such as an antibody, which gives the correspondingly high therapeutic selectivity. FA functionalization enables efficient and rapid transfer of the therapeutic agent via FR1-mediated endocytosis [11]. The importance of the localization studies performed here is also illustrated by the fact that pure protein quantification could not show a clear relationship between FR1 expression and, e.g., overall survival of PCa patients. For instance, the TCGA database indicates a potential elevation in overall survival rates with higher FOLR1 levels (*p* = 0.064) [46]. Rather, the altered activation and localization in PCa seem to represent a clinically significant analysis, which was established here.

This method can both mature into a therapeutic target and represent diagnostic and prognostic strategies that, through modern technologies such as AI-based image analysis, can lead to a significant improvement in the treatment of our patients.

## 5. Conclusions

FR1 shows altered processing and expression in prostate cancer compared to other neoplasms. Despite low mRNA transcripts, image analysis demonstrated a distinct modified localization through activation of GPI-T, as well as functional localization of the receptor in the cell membrane, shown by its subsequent functional use as a therapeutic target receptor. Thus, in addition to the therapeutic potential of FR1, this study also demonstrates a high diagnostic potential of this particular receptor, which could potentially allow the differentiation of biological tumor subtypes and risk stratifications based on tumor samples.

## Figures and Tables

**Figure 1 cancers-16-02008-f001:**
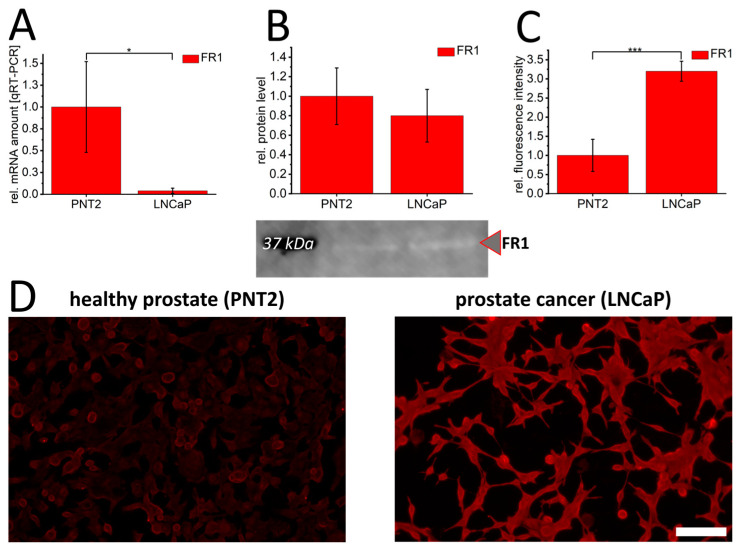
Folate receptor 1 expression in healthy PNT2 and prostate cancer cells (LNCaP). Folate receptor 1 (FR1/red) was quantified by qRT-PCR (**A**), Western blot (**B**) and immunofluorescence (**C**,**D**). The graphs show relative values calculated to healthy control cells. In (**B**), a representative Western blot membrane with the corresponding marker band (black) at 37 kDa is shown. The effective protein load was detected by BioRad-stain-free technology, enabling adjustment of protein loads by quantification of total protein levels. *p*-values of * 0.05, ** 0.01 and *** 0.001 were labeled with one to three asterisks indicating significances from at least 3 independent experiments. Scalebar 100 µm.

**Figure 2 cancers-16-02008-f002:**
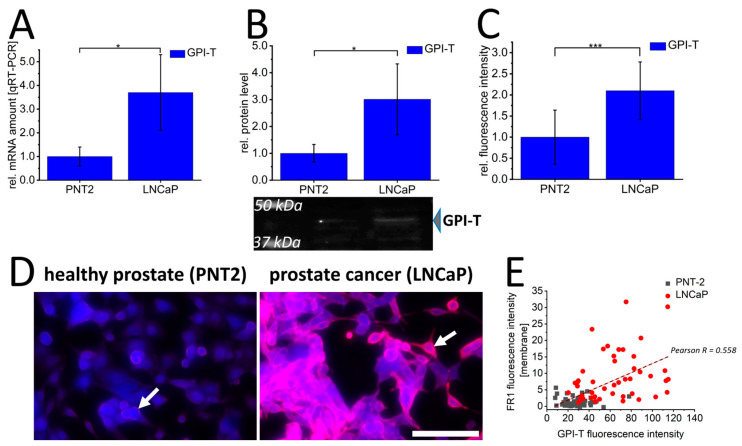
Glycosylphosphatidylinositol transamidase expression and folate receptor 1 localization in prostate cancer cells (LNCaP) and healthy prostate epithelial cells (PNT2). Glycosylphosphatidylinositol transamidase (GPI-T/blue) was quantified by qRT-PCR (**A**), Western blot (**B**) and immunofluorescence (**C**,**D**) in healthy PNT2 and prostate cancer cells (LNCaP). In (**D**), immunofluorescence against FR1 (red) is additionally shown; white arrows indicate membrane signals. Correlation between FR1-membrane fluorescence intensity and GPI-T fluorescence intensity in healthy PNT2 (gray squares) and LNCaP cancer cells (red circles) with corresponding correlation value Pearson R is shown in (**E**). The graphs (**A**–**C**) show the relative values calculated for the healthy control cell. (**B**) shows a representative Western blot membrane with the corresponding marker bands at 37 kDa and 50 kDa. The effective protein load was detected by BioRad-stain-free technology, enabling adjustment of protein loads by quantification of total protein levels. *p*-values of * 0.05, ** 0.01 and *** 0.001 were labeled with one to three asterisks indicating significances from at least 3 independent experiments. Scalebar 100 µm.

**Figure 3 cancers-16-02008-f003:**
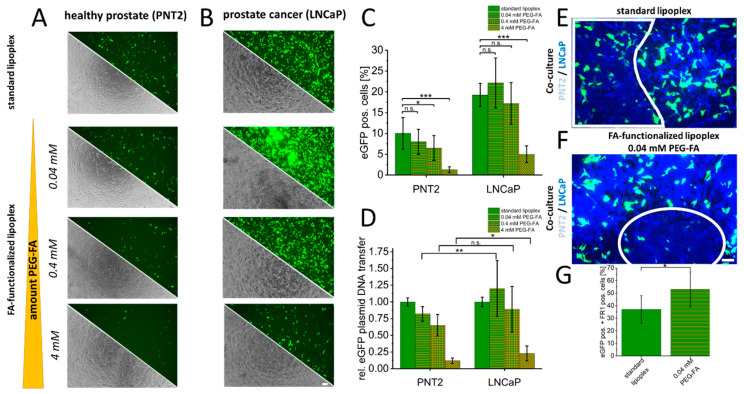
Folate-functionalized lipoplexes for selective transfer of eGFP-plasmid DNA into cancer cells. As a control, eGFP plasmid DNA was transferred into healthy PNT2 and LNCaP cancer cells using unmodified standard lipoplexes. In addition, an increasing concentration of NHS-PEG-FA was used to characterize specific uptake via FR1 examined by eGFP-positive cells using microscopy (**A**,**B**). In addition, quantification was performed by flow cytometry, which provided information on the changes in transfection efficiency for each cell type (**C**) and the altered specificity between FR1-mediated uptake in PNT2 and LNCaP (**D**). Finally, in co-culture, this specific transfection of standard lipoplexes (**E**) and FA-functionalized lipoplexes (**F**) was further investigated. The healthy cells are shown demarcated in the white frames and could be identified by GPI-T immunofluorescence (weak blue) based on the lower fluorescence intensity compared to LNCaP cells (bright blue). Quantification of enhanced selectivity in co-culture is shown in (**G**). *p*-values of * 0.05, ** 0.01 and *** 0.001 were labeled with one to three asterisks indicating significances from at least 3 independent experiments. n.s., not significant. Scale bar 100 µm.

**Figure 4 cancers-16-02008-f004:**
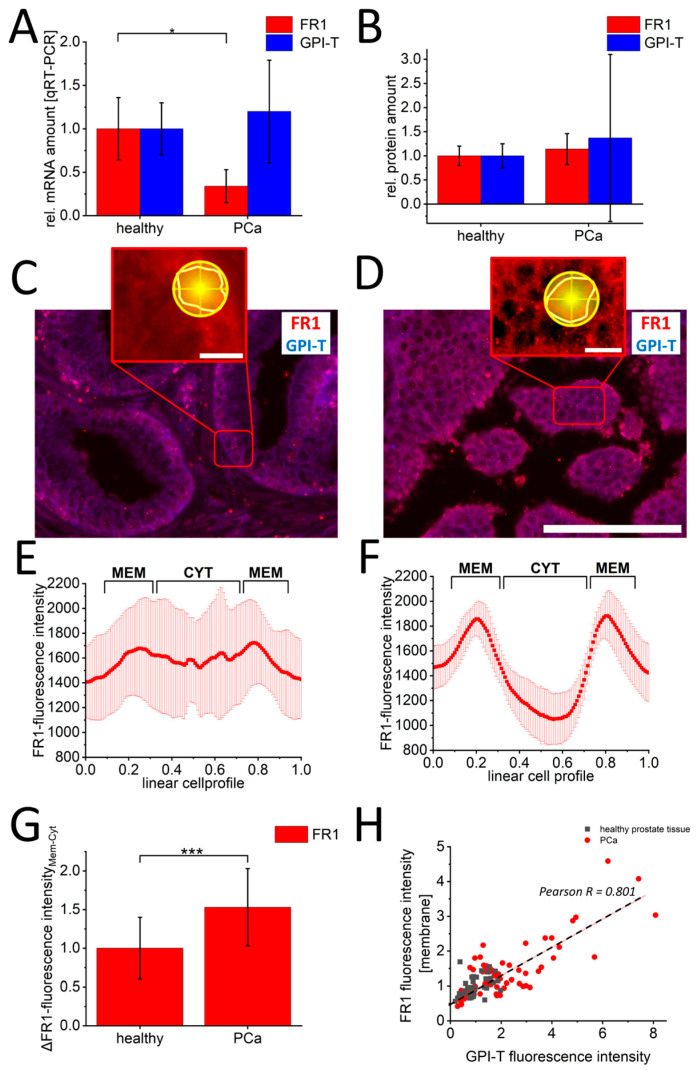
Characterization of GPI-T and folate receptor-1 in prostate cancer and healthy prostate tissue. Quantification of FR1 (red) and GPI-T (blue) mRNA by qRT-PCR is shown in (**A**), and protein by Western blot is shown in (**B**). In (**C**,**D**) representative images of immunofluorescence against FR1 and GPI-T are shown. The red boxes show a zoom-in to illustrate the analysis procedure. For this purpose, cells in the DAPI channel were randomly selected. After zooming into the FR1 channel, cells were marked with a polygon. The following line quantification was performed with 180 lines (yellow circle shape). The corresponding fluorescence profiles of healthy (**E**) and malignant (**F**) cells are shown exemplarily. The signal shift of FR1 signal from cell cytoplasm to cell membrane shows a significant change in PCa sections ((**G**), *n* > 300 healthy cells, *n* > 600 PCa cells). This change shows a positive correlation (Pearson R = 0.801) to the increase in GPI-T signals (**H**). Scalebar 100 µm (**C**,**D**) and 10 µm (zoom-in images). Measured values from *n* = 3 healthy and *n* = 6 PCa sections. *p*-values of * 0.05, ** 0.01 and *** 0.001 were labeled with one to three asterisks indicating significances.

**Table 1 cancers-16-02008-t001:** The composition of RIPA lysis buffer used for protein isolation of cell cultures.

Buffer	Content	Manufacturer (Order ID.)	Amount
RIPA lysis buffer	TRIS HCl	Carl Roth, Karlsruhe, Germany (9090.2)	2.42 g
	NaCl	Carl Roth (3957.1)	8.76 g
	Nonidet P 40-Replacement product solution	Merck (74388)	20 mL
	SDS pellets	Carl Roth (CN30.2)	1 g
	Deoxycholic acid sodium salt	Carl Roth (3484.2)	5 g
	Demineralized water		Fill to 1 L
RIPA lysis buffer + inhibitors	Protease inhibitor	Merck (11873580001)	1/2 tablet
	Phosphatase inhibitor	Merck (P0044)	50 µL
	RIPA lysis buffer		Fill to 5 mL

**Table 2 cancers-16-02008-t002:** Summary of all used primary and secondary antibodies with corresponding methods and used dilutions: WB = Western blot; IF-c = immunofluorescence cell culture; IF-t = immunofluorescence tissue; IF-c-t = immunofluorescence cell culture after transfection. The font color shows the representation of the respective antibodies throughout all figures. The green asterisk *****: indicates the antibody specifications used for Occludin.

	Secondary Antibody	StarBright Blue 700 Goat Anti-Rabbit IgG (Bio-Rad, 12004161)	StarBright Blue 520 Goat Anti-Mouse IgG (Bio-Rad, 12005866)	Goat-Anti Rabbit IgG Alexa Fluor 488 (Abcam, ab150116, Cambridge, UK)	Goat-Anti Mouse IgG Alexa Fluor 594 (Abcam, ab150077)	Goat Anti-Rabbit IgG Alexa Fluor 594 (Thermo Scientific, A-11012)
Primary Antibody						
**GPI-T**—PIGK (N-Term) (antikörper-online, ABIN389064) **Occludin (invitrogen, 1529359A)**		WB (Section 2.4.5): 1:10,000	-	IF-c (Section 2.4.6) *****: 1:400 IF-t (Section 2.4.6): 1:400	-	IF-c-t (Section 2.5.3): 1:400
		WB (Section 2.4.5): 1:1000		IF-c (Section 2.4.6) *****: 1:100 IF-t (Section 2.4.6): 1:50		IF-c-t (Section 2.5.3): 1:100
**FR1**—FOLR1 (AA 41-227) (antikörper-online, ABIN5611335)		-	WB (Section 2.4.5): 1:10,000	-	IF-c (Section 2.4.6): 1:400 IF-t (Section 2.4.6): 1:400	-
			WB (Section 2.4.5): 1:1000		IF-c (Section 2.4.6): 1:100 IF-t (Section 2.4.6): 1:50	

**Table 3 cancers-16-02008-t003:** Microscope settings for immunofluorescence and transfection experiments.

Experiment	Channel	Magnification	Exposure Time [ms]	Gain
Immunofluorescence of cell culture (Section 2.4.6)	GFP	40×/20×	150	10
	RFP	40×/20×	150	10
	DAPI	40×	10	10
Immunofluorescence of tissue (Section 2.4.6)	GFP	40×	538	10
	RFP	40×	4180	30
	DAPI	40×	513	10
Transfection (Section 2.5.1/Section 2.5.2)	GFP	4×	18	50
Transfection and immunofluorescence staining (Section 2.5.3)	GFP	10×	18	37
	RFP	10×	164	89

## Data Availability

The data that support the findings of this study are available from the corresponding author upon reasonable request.

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
