# Peer review of "Therapeutic and Diagnostic Potential of Folic Acid Receptors and Glycosylphosphatidylinositol (GPI) Transamidase in Prostate Cancer"

_cancers, 2024, doi:10.3390/cancers16112008_

Round 1

Reviewer 1 Report (Previous Reviewer 3)

Comments and Suggestions for Authors

Authors responded all the comments appropriately. 

Reviewer 2 Report (Previous Reviewer 4)

Comments and Suggestions for Authors

The authors responded to the critical comment appropriately.

This manuscript is a resubmission of an earlier submission. The following is a list of the peer review reports and author responses from that submission.

Round 1

Reviewer 1 Report

Comments and Suggestions for Authors

This is an excellent paper with carefully done studies in prostate tissues. However, whereas PSMA is relatively specific for prostate tissue and salivary tissue, this is not the case for FR1. It is found in most tissues. A multi-tissue blot should be shown and staining performed with each of the methods use in the current version. Without this data proposed use of FR1 such as for identification of prostate cancer vs normal is not possible. More important is the use of FR1 as a targeting agent for therapies. If expressing in multiple tissues, toxicity would be a problem. To some degree this is already already a problem with use of PSMA as a targeting agent.

Comments on the Quality of English Language

none

Author Response

We deeply appreciate your supportive feedback, which has greatly enriched the quality of our manuscript. Below, we will succinctly address the key points and endeavor to provide satisfactory responses to your inquiries: We fully agree with your assertion. The literature and various proteome analyses extensively document the ubiquitous expression of FR1 across tissues, primarily attributed to its role as a transporter for folic acids. We have thoroughly discussed these findings, along with their implications and limitations, in both the introduction and discussion sections of our manuscript. Although corresponding quantifications of protein quantities have already been described many times and have also been correlated with clinical courses in the TCGA database, for example, we have so far underscored the significance and the comparison with our results.  Consequently, we have included a dedicated section within our discussion that focuses on the limitations and translational significance of our results (page 16, line 602-615).   Moreover, the utilization of FR1 as a target receptor, exemplified by MORAb-202 mentioned in our introduction, serves as a preliminary screening rather than the sole determinant in the selectivity of the therapeutic agent. This mechanism ensures swift endocytic uptake by target cells, with subsequent selection facilitated by other components such as an effective coupled antibody, ensuring ultimate specificity and selectivity. We have further explored this aspect within the discussion section, elaborating on the limitations of FR1 (page 16, line 606-609).

Reviewer 2 Report

Comments and Suggestions for Authors

v                 Suggestions to Author/s

  1. Dear Dr. Hoffmann, as a selected reviewer, I made a prompt chek of your excellent article entiteled: “Therapeutic and diagnostic potential of Folic acid receptors and  Glycosylphosphatidylinositol (GPI) Transamidase in Prostate Cancer “and found it: (x) Excellent, accept the submission (5).
  2. During the prompt check, some small mistakes were found. The text was converted

from pdf in original to doc by the use of Convertio software and in doc form it was corrected. The corrected words were highlited in red color. Please accept them and highlited them into black back. After, please by the use of Convertio convert back the text into pdf.

  1. Important finding, the additional language and editing is not needed according to my opinion.

Author Response

We extend our sincere gratitude for your diligent efforts in thoroughly proofreading our manuscript. We have taken your suggestion into account and made appropriate adjustments in track changes mode.

Reviewer 3 Report

Comments and Suggestions for Authors

The present study demonstrated the therapeutic and diagnostic potential of folic acid receptors and glycosylphosphatidylinositol (GPI) transamidase in prostate cancer. The authors adequately carried out and analyzed the results. However, the manuscript is good, and the authors should address the following comment:.

1. Authors should show beta-action or GAPDH for western blot images (figures 1B and 2B).

2. Figure 1B Western blot and immunofluorescence images show FR1 expression is high in LnCap cells as compared to PNT1, but in the abstract, authors show FR1 expression is high in PNT1 as compared to LnCap Explain.

3. Figure 3A, 2D, 4C, and D immunofluorescence images are not clear. Explain.

4. The authors showed that FR1 has high diagnostic potential for prostate cancer. If authors show immunohistomical expression of FR1 in prostate cancer patients, it would be more beneficial for this manuscript.

Author Response

We would also like to thank you very much for your efforts and exceptionally well-structured comments on our manuscript. Your supportive feedback has contributed to optimizing our manuscript. Below, we briefly discuss the main points and aim to address your concerns satisfactorily:

  1. You are correct that we did not include these comparisons in the Western blot. As stated in the material and methods section, we utilized the new BioRad stain-free system, allowing for the detection of actual protein amounts transferred to the membrane. We adjusted accordingly based on this total protein signal. To clarify and directly reference this procedure, we have added this information beneath the Western blot images (page 10, line 388-389, page 10, line 414-415) .
  2. Presumably you are aiming at the different mRNA and resulting protein expression of these two cell types. This is indeed very surprising. However, we have already tried to shed light on these aspects in the discussion and provided possible explanations, such as the function as a transcription factor or the rapid availability with a higher FR requirement in healthy prostates. Further translational silencing processes cannot be ruled out here. However, as our focus was primarily on the differential localization of FR1 in healthy and cancer cells, we did not pursue further experiments to investigate this discrepancy. Corresponding assumptions about this phenomenon are considered in the discussion on page 14 from line 510 - 519 and summarize the possible regulatory mechanisms for this discrepancy between mRNA and protein quantity.
  3. The slight lack of clarity in the figures may stem from the conversion to PDF format. All images were thoroughly checked for quality and signal-to-noise ratio, ensuring sufficient clarity for the intended investigation (localization study). By submitting the images separately in high resolution and subsequently integrating them into the manuscript, we ensure a high-resolution version upon publication.
  4. Due to our chosen evaluation method and the localization studies as well as co-staining with GPI-T and FR1, we opted for immunofluorescence (IF). This is the only way to ensure the single-cell-based, high-resolution simultaneous analysis of the FR1 and GPI-T proteins and the corresponding localization of the signals. While immune-histo-chemistry (IHC) could offer simpler quantification, it is unfortunately not suitable for our focus on detailed protein localization.

Reviewer 4 Report

Comments and Suggestions for Authors

This manuscript discusses the potential role of the folate receptor (FOLR1) in prostate cancer. While multiple papers touch upon FOLR1, its specific impact on prostate cancer remains uncertain. This study operates on an underlying assumption that requires validation before exploring its therapeutic and diagnostic potential.

Major points:

Expression of FOLR1: Analysis of TCGA data reveals that in prostate adenocarcinoma, FOLR1 transcript levels are higher in normal tissues compared to cancerous tissues. The authors must demonstrate the significance of FOLR1 as a contributor to prostate cancer progression.

Survival Rate: A higher FOLR1 transcript level doesn't significantly impact overall survival rates. In prostate cancer, the analysis shows no statistically significant difference. Notably, however, there's a suggestive trend (p = 0.064) indicating a potential elevation in overall survival rates with higher FOLR1 levels in TCGA data. This observation contradicts the study's initial justification, raising the question of why the authors focused their analysis on FOLR1 in the first place.

Comments on the Quality of English Language

The overall English qaulity is good.

Author Response

Thank you for these important and relevant insights during the review of our manuscript. You raise an important point that reinforces the significance of our study. The TCGA database examines total FR protein levels and their correlation with overall survival. As we have also demonstrated, total protein values from Western blot analysis do not significantly differ, making it challenging to distinguish between healthy and diseased prostates through simple quantification of FR signals. Instead, our focus on localization and activation, independent of underlying FR protein concentrations, enables detailed single-cell analysis.  To actively incorporate this point and your justified comment, we have included corresponding TCGA database data in the discussion section, aiming to substantiate the significance of our study and clarify the crucial difference between pure protein quantity and localization in prostate carcinoma.